# Towards Principled Objectives for Contrastive Disentanglement

## Abstract

Unsupervised learning is an important tool that has received a significant amount of attention for decades. Its goal is 'unsupervised recovery,' i.e., extracting salient factors/properties from unlabeled data. Because of the challenges in defining salient properties, recently, 'contrastive disentanglement' has gained popularity to discover the additional variations that are enhanced in one dataset relative to another. Existing formulations have devised a variety of losses for this task. However, all present day methods exhibit two major shortcomings: (1) encodings for data that does not exhibit salient factors are not pushed to carry no signal; and (2) introduced losses are often hard to estimate and require additional trainable parameters. We present a new formulation for contrastive disentanglement which avoids both shortcomings by carefully formulating a probabilistic model and by using non-parametric yet easily computable metrics. We show on four challenging datasets that the proposed approach is able to better disentangle salient factors.

## 1 Introduction

Unsupervised machine learning requires to extract latent information from unlabeled data points. For instance, given a dataset of digits we want data points to be grouped according to the depicted number, e.g., by clustering into a set of groups. Many algorithms have been developed and commonly their goal is 'unsupervised recovery,' i.e., to *extract salient factors/properties* from unlabeled data.

However, unsupervised recovery is inherently challenging because saliency of a factor/property/feature can be hard to define and in case of visual data, can depend on the viewpoint. Recent work (Abid et al., 2017; Severson et al., 2018; Ruiz et al., 2019) has therefore advocated for 'contrastive disentanglement': Given two sets of data without obvious correspondence between their members, *discover the additional variations* that are enhanced in one dataset relative to another. For instance, one dataset may depict grass while the other illustrates grass with digits overlaid. In this case the digits are the additional variations/enhancements that need to be disentangled. This contrastive setting is useful across many important applications in unsupervised learning, e.g., when comparing to control groups or when disentangling variations from data.

For this contrastive setting a set of techniques have been proposed very recently to disentangle variations (Abid et al., 2017; Severson et al., 2018; Ruiz et al., 2019). Common to all those techniques is the idea that observed data is modeled as the linear or non-linear combination of two transformed latent signals. For example, one latent signal enables reconstruction of one entangled dataset (e.g., grass dataset), while a second latent signal models the 'additional variations' of the second dataset when compared to the first (e.g., the digits). Furthermore, these methods introduce additional losses in order to better disentangle background factors from salient features. While those losses were shown to improve results, they exhibit two key shortcomings: (1) the formulation ignores information about the prior, i.e., the salient features of the background data should be zero; and (2) additional KL-divergence-based losses which are hard to estimate in practice are introduced to improve disentanglement. However, reasonably accurate estimation of those losses requires additional trainable parameters, which increases the model size.

To fix those two shortcomings we instead propose two new losses for disentanglement. The first actively encourages the salient factors to be zero if additional variations are not available. We show that this loss originates from a careful probabilistic derivation of the disentanglement setting. The second *non-parametric* loss encourages the background distributions to be identical. Different from

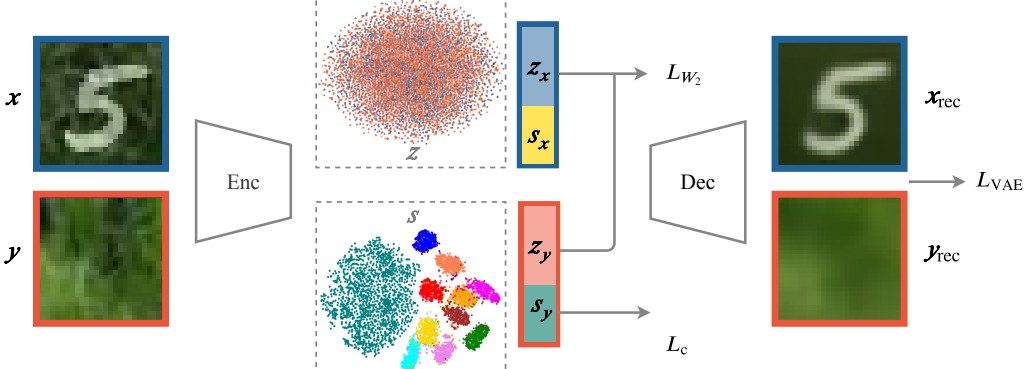

Figure 1: Our VAE framework to achieve contrastive disentanglement. We propose the two new losses $L_{W_2}$ and $L_C$ to improve disentanglement. Clustering in $z$ space has overlapping clusters because the common factors are encouraged to be distributionally similar. Clustering in the $s$ space has clearly distinct clusters representing target, background, and intra-target classes.

existing methods we propose to use a *non-parametric* loss which is easy to compute and does not require additional trainable parameters.

We demonstrate our contributed loss functions on a variety of datasets as highlighted in Table 1. We validate through a set of challenging qualitative and quantitative benchmarks that our first new loss gives rise to a pair of disentangled latent features. Further, we empirically establish that our second new loss encourages the common factors of the target and background data to have identical distributions, while baseline losses do not exhibit a similar behavior.

## 2 BACKGROUND

In this section, we formally state the contrastive disentanglement objective and recall existing approaches to address it as well as their shortcomings. Let $\{x^{(i)}\}_{i=1}^n$ and $\{y^{(j)}\}_{j=1}^m$ be two unlabelled datasets with each $x^{(i)}, y^{(j)} \in \mathbb{R}^d$. We refer to $\{x^{(i)}\}$ as the target dataset, whereas $\{y^{(j)}\}$ is the background dataset. We assume that the samples $\{x^{(i)}\}$ are drawn i.i.d. from a target distribution $p_t(x)$, whereas $\{y^{(j)}\}$ are drawn i.i.d. from a background distribution $p_b(y)$. For example, in the Grassy-MNIST dataset (detailed in Section 4), the target dataset $\{x^{(i)}\}$ is the set of images with MNIST digits superimposing image patches showing grass, and the background set $\{y^{(j)}\}$ is comprised of images showing only grass. The target dataset exhibits features that have commonality with the features of the background data, e.g., grass, in addition to some unique interesting features that are specific only to the target data, e.g., digits. Thus the primary objective in the contrastive disentanglement setting is the following:

**Goal:** Contrastive disentanglement aims to extract a pair of disentangled latent features $(s, z)$ such that the salient feature $s$ encodes the generative factors of interest, e.g., digits, that are enhanced in $\{x^{(i)}\}$ relative to $\{y^{(j)}\}$, whereas the background feature $z$ explains the common sources of variation in these two datasets, e.g., grass background.

A key aspect of this setting is that the salient feature $s$ is constant across the background dataset $\{y^{(j)}\}$ and is enhanced only in the target set $\{x^{(j)}\}$. However, apart from the unlabelled datasets of images, no additional information about these generative factors of interest, such as the background-target pairs nor target images with a specific salient feature are provided. Thus the key challenge is to exploit this implicit information inherent to the structure of the two datasets to learn disentangled features. We now review some relevant approaches to learn these disentangled factors.

**VAE.** Given training samples generated from a single unknown distribution $p(x)$, a variational auto-encoder (VAE) models the joint distribution of the observation and the latent space through parametric deep nets $p_\theta(x, z)$. Hereby $z$ is the lower dimensional latent vector and $x$ is the high dimensional observation, such as an image. The joint distribution factorize into $p_\theta(x, z) = p(z)p_\theta(x|z)$, where the prior $p(z)$ is assumed to be a standard Gaussian and the conditional $p_\theta(x|z)$ is assumed to be parameterized by a deep net referred to as the 'decoder.' Since the posterior $p_\theta(z|x)$ is hard to compute, VAEs approximate this posterior with a parameterized deep net $q_\phi(z|x)$, called the 'encoder.' In particular, $q_\phi(z|x)$ too is assumed to be a factored Gaussian with mean and diagonal

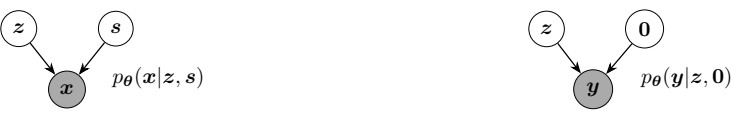

(a) Target latent model  (b) Background latent model

Figure 2: Latent variable model for target and background. Shaded circles represent observed variables.

covariance determined by the encoder deep net. For training, the encoder and decoder parameters $\phi$ and $\theta$ are learnt by minimizing the objective function $L(\theta, \phi)$, given by

$$L(\theta, \phi) = \mathbb{E}_{p(\boldsymbol{x})}[D_{\mathrm{KL}}(q_\phi(\boldsymbol{z}|\boldsymbol{x}) \| p(\boldsymbol{z})) - \mathbb{E}_{q_\phi(\boldsymbol{z}|\boldsymbol{x})} \log p_\theta(\boldsymbol{x}|\boldsymbol{z})], \tag{1}$$

where $L(\theta, \phi)$ serves as an upper bound to the negative log-likelihood of the data.

This classical VAE framework for a single dataset can be extended to the contrastive setting which involves multiple datasets (Ruiz et al., 2019; Abid & Zou, 2019). The main idea is that the samples generated from the two unknown distributions, i.e., the target distribution $p_t(\boldsymbol{x})$ and the background distribution $p_b(\boldsymbol{y})$, can be captured by a pair of independent generative latent factors $(\boldsymbol{z}, \boldsymbol{s})$. Here $\boldsymbol{z}$ is thought of as a 'common' factor that captures variations in samples from both the distributions, whereas $\boldsymbol{s}$ is the 'salient' factor that encodes generative factors of interest that are unique only to the target samples. Thus the generative model for the target dataset is modeled by a conditional Gaussian distribution, i.e., $\boldsymbol{x}_i \sim p_\theta(\boldsymbol{x}|\boldsymbol{z}_i, \boldsymbol{s}_i)$, where $\theta$ denotes the set of decoder parameters. Since the common factor $\boldsymbol{z}$ alone encodes the background dataset, we model it as $\boldsymbol{y}_j \sim p_\theta(\boldsymbol{y}|\boldsymbol{z}_j, \boldsymbol{s}_j = \boldsymbol{0})$, where we set the constant salient factor to zero, without loss of generality. Further, similar to the VAE setup above, we approximate the posteriors over the latent features by a factored conditional Gaussian distribution, i.e., $(\boldsymbol{z}_i, \boldsymbol{s}_i)|\boldsymbol{x}_i \sim q_\phi(\boldsymbol{z}|\boldsymbol{x}_i)q_\phi(\boldsymbol{s}|\boldsymbol{x}_i)$ and $(\boldsymbol{z}_j, \boldsymbol{s}_j)|\boldsymbol{y}_j \sim q_\phi(\boldsymbol{z}|\boldsymbol{y}_j)q_\phi(\boldsymbol{s}|\boldsymbol{y}_j)$, where $\phi$ denotes the set of encoder parameters which capture and the mean and diagonal variances of these approximate posterior distributions. Note that both datasets share the same encoder and decoder.

Another key component of the generative model is the latent prior. Since both of the independent latent factors $\boldsymbol{z}$ and $\boldsymbol{s}$ encode the target samples, we define the target latent prior $p_t(\boldsymbol{z}, \boldsymbol{s})$ as $p_t(\boldsymbol{z}, \boldsymbol{s}) \triangleq p_t(\boldsymbol{z})p_t(\boldsymbol{s}) = \mathcal{N}(\boldsymbol{z}; \boldsymbol{0}, \boldsymbol{I}_{d_z}) \cdot \mathcal{N}(\boldsymbol{s}; \boldsymbol{0}, \boldsymbol{I}_{d_s})$, where $\mathcal{N}(\boldsymbol{z}; \boldsymbol{0}, \boldsymbol{I}_{d_z})$ denotes a standard Gaussian distribution in $d_{\boldsymbol{z}}$-dimensional Euclidean space, and $\mathcal{N}(\boldsymbol{s}; \boldsymbol{0}, \boldsymbol{I}_{d_s})$ is defined similarly. Furthermore, to accommodate the special structure of the contrastive disentanglement setting, i.e., the background $\{\boldsymbol{y}^{(j)}\}$ is captured by $\boldsymbol{z}$ alone with $\boldsymbol{s}$ being a constant, we define the background latent prior $p_b(\boldsymbol{z}, \boldsymbol{s})$ as

$$p_b(\boldsymbol{z}, \boldsymbol{s}) \triangleq p_b(\boldsymbol{z})p_b(\boldsymbol{s}) = \mathcal{N}(\boldsymbol{z}; \boldsymbol{0}, \boldsymbol{I}_{d_z}) \cdot \delta\{\boldsymbol{s} = \boldsymbol{0}\}, \tag{2}$$

where $\delta\{\boldsymbol{s} = \boldsymbol{0}\}$ denotes the Dirac distribution centred at zero. Figure 2 illustrates this latent variable model. Using this latent variable model setup, Ruiz et al. (2019) and Abid & Zou (2019) derive the following loss $L_{\mathrm{VAE}}$ as an upper bound for the sum of negative log-likelihood of both the datasets:

$$L_{\mathrm{VAE}} = \mathbb{E}_{p_t(\boldsymbol{x})}[D_{\mathrm{KL}}(q_\phi(\boldsymbol{z}|\boldsymbol{x})q_\phi(\boldsymbol{s}|\boldsymbol{x}) \| p_t(\boldsymbol{z})p_t(\boldsymbol{s})) - \mathbb{E}_{q_\phi(\boldsymbol{z}|\boldsymbol{x})q_\phi(\boldsymbol{s}|\boldsymbol{x})} \log p_\theta(\boldsymbol{x}|\boldsymbol{z}, \boldsymbol{s})] + $$
$$\mathbb{E}_{p_b(\boldsymbol{y})}[D_{\mathrm{KL}}(q_\phi(\boldsymbol{z}|\boldsymbol{y}) \| p_b(\boldsymbol{z})) - \mathbb{E}_{q_\phi(\boldsymbol{z}|\boldsymbol{y})} \log p_\theta(\boldsymbol{y}|\boldsymbol{z}, \boldsymbol{s} = 0)]. \tag{3}$$

This classical VAE loss $L_{\mathrm{VAE}}$ defined above is the basis for cVAE and sRbVAE which we discuss below. However, we already want to highlight an important observation regarding the loss in Eq. (3): by using the above latent variable model for the contrastive setup and plugging them in Eq. (1), the actual combined negative log-likelihood is bounded by a loss that contains a crucial KL divergence term between the posterior $q_\phi(\boldsymbol{s}|\boldsymbol{y})$ and the Dirac latent prior $p_b(\boldsymbol{s}) = \delta\{\boldsymbol{s} = \boldsymbol{0}\}$ in addition to the $L_{\mathrm{VAE}}$ defined above. We elaborate on this further in Section 3.

**cVAE.** In a recent work, Abid & Zou (2019) introduced the Contrastive Variational Auto Encoder (cVAE) for learning disentangled latent features. In particular, they consider the same latent variable model setup as above. In addition, to further encourage the disentanglement between the latent factors $\boldsymbol{z}$ and $\boldsymbol{s}$, they add the following total correlation (TC) term to above objective in Eq. (3):

$$L_{\mathrm{TC}} = \mathbb{E}_{p(\boldsymbol{x})}[D_{\mathrm{KL}}(\bar{q} \| q_\phi(\boldsymbol{z}|\boldsymbol{x})q_\phi(\boldsymbol{s}|\boldsymbol{x}))],$$

where $\bar{q} \triangleq q_\phi(\boldsymbol{z}, \boldsymbol{s}|\boldsymbol{x}_i)$ denotes the joint conditional probability of the latent features. The overall objective for the cVAE framework is thus given by:

$$\min_{(\theta, \phi)} L_{\mathrm{cVAE}} \triangleq L_{\mathrm{VAE}} + \lambda \cdot L_{\mathrm{TC}}, \tag{4}$$

where $\lambda > 0$ is a hyperparameter.

**sRb-VAE.** Recently, Ruiz et al. (2019) introduced the Symmetric Reference-based Variational Auto Encoder (sRb-VAE) for learning the disentangled features $(\boldsymbol{s}, \boldsymbol{z})$. Utilizing the fact that minimizing the loss $L_{\text{VAE}}$ in Eq. (3) is equivalent to minimizing the KL divergence over the joint distributions $D_{\text{KL}}(q_\phi(\boldsymbol{x}, \boldsymbol{z}, \boldsymbol{s}, c) \parallel p_\theta(\boldsymbol{x}, \boldsymbol{z}, \boldsymbol{s}, c))$, where $c \in \{0, 1\}$ indicates if $\boldsymbol{x}$ is either from the background or target dataset, the authors add an additional KL divergence term to make the overall objective symmetric, i.e.,

$$\min_{(\boldsymbol{\theta}, \boldsymbol{\phi})} \ L_{\text{sRb-VAE}} \triangleq L_{\text{VAE}} + D_{\text{KL}}(p_\theta(\boldsymbol{x}, \boldsymbol{z}, \boldsymbol{s}, c) \parallel q_\phi(\boldsymbol{x}, \boldsymbol{z}, \boldsymbol{s}, c)). \tag{5}$$

Note that the distribution $q_\phi(\boldsymbol{x}, \boldsymbol{z}, \boldsymbol{s}, c)$ is defined as $q_\phi(\boldsymbol{x}, \boldsymbol{z}, \boldsymbol{s}, c) \triangleq p(c)p(\boldsymbol{x}|c)q_\phi(\boldsymbol{z}|\boldsymbol{x})q_\phi(\boldsymbol{s}|\boldsymbol{x})$ where $p(c = 0) = p(c = 1) = 1/2$ and $p(\boldsymbol{x}|c = 1) = p_t(\boldsymbol{x})$, and $p(\boldsymbol{x}|c = 0) = p_b(\boldsymbol{x})$ denotes the target and background distributions respectively. Similarly, $p_\theta(\boldsymbol{x}, \boldsymbol{z}, \boldsymbol{s}, c) \triangleq p(c)p(\boldsymbol{z}, \boldsymbol{s}|c)p_\theta(\boldsymbol{x}|\boldsymbol{z}, \boldsymbol{s})$, where $p(\boldsymbol{z}, \boldsymbol{s}|c = 1) = p_t(\boldsymbol{z}, \boldsymbol{s})$ and $p(\boldsymbol{z}, \boldsymbol{s}|c = 0) = p_b(\boldsymbol{z}, \boldsymbol{s})$ denote the target and background latent priors.

**Shortcomings:** Since both cVAE and sRb-VAE build upon the VAE framework, their losses involve the same $L_{\text{VAE}}$ defined in Eq. (3). However, this VAE loss is not completely reflective of the assumptions about the background latent prior $p_b(\boldsymbol{z}, \boldsymbol{s})$ defined in Eq. (2). In particular, due to the absence of the KL divergence term $D_{\text{KL}}(q_\phi(\boldsymbol{s}|\boldsymbol{y}) \parallel \delta\{\boldsymbol{s} = \boldsymbol{0}\})$ in Eq. (3), the existing objective $L_{\text{VAE}}$ does not enforce the salient latent feature $\boldsymbol{s}$ to be zero for the background dataset, which is contrary to our modeling assumptions. Intuitively, by more strictly encouraging the salient latent feature to be zero, we will prevent information from being encoded in this vector. In addition, to encouraging disentanglement, both aforementioned approaches add new KL divergence based losses to the standard objective. However these new divergence terms involve estimation of ratios of densities which are quite hard to approximate. Hence they utilize the classical trick of discriminator neural nets (Kim & Mnih, 2018). This makes the overall training procedure more complex since this discriminator should approximate this ratio at each training step of the encoder parameters $(\boldsymbol{\theta}, \boldsymbol{\phi})$. This also increases the number of trainable parameters together with an added difficulty of finding the right discriminator architecture to estimate this ratio. In contrast, in this paper, we provide a simple and principled approach to promote disentanglement that performs at least as good as both of these methods on a variety of datasets (see Section 4). In addition, our architectural complexity is identical to a classical VAE as opposed to the complex architectures of existing approaches.

## 3 APPROACH

In this section, we address the aforementioned shortcomings and present our novel approach towards the desired goal of contrastive disentanglement. In particular, we introduce two novel loss terms that are reflective of the structure inherent to the contrastive setting and that help attain disentangled representations. Whereas the first loss term enforces the information about the common background features to be encoded in $\boldsymbol{z}$ alone, the second loss term encourages the distribution of the common factors to be the same across both the datasets. Figure 1 illustrates our approach. We now discuss details about each of these loss terms below.

### 3.1 CORRECT VAE LOSS

To gain intuition for our approach, recall the central aim of contrastive disentanglement: (1) encode information about the common features within both datasets in the common factor $\boldsymbol{z}$ alone; and (2) encode salient features of interest, that are unique to the target samples, in just $\boldsymbol{s}$. While this is incorporated in the background and target latent priors, the classical VAE loss $L_{\text{VAE}}$ given in Eq. (3), and used by Abid & Zou (2019); Ruiz et al. (2019), does not capitalize on this. To see this, note that while the reconstruction error term $\mathbb{E}_{q_\phi(\boldsymbol{z}|\boldsymbol{y})} \log p_\theta(\boldsymbol{y}|\boldsymbol{z}, \boldsymbol{s} = 0)$ enforces the decoder to reconstruct the background image $\boldsymbol{y}$ from just $\boldsymbol{z}$ with $\boldsymbol{s}$ being zero, the KL divergence loss $D_{\text{KL}}(q_\phi(\boldsymbol{z}|\boldsymbol{y}) \parallel p_b(\boldsymbol{z}))$ does not enforce the information about the background to be encoded in only $\boldsymbol{z}$ since it does not penalize any leakage of the background features into the salient feature $\boldsymbol{s}$. To address this issue in a principled manner, we propose to add a new loss term that exploits the background latent prior from Eq. (2) and that achieves our desired goal of disentangled representations.

We now derive our new loss mathematically. Denoting the respective negative log-likelihoods of the background and target data as $\mathcal{L}(p_\theta(\boldsymbol{y})) \triangleq \mathbb{E}_{p_b(\boldsymbol{y})}[-\log p_\theta(\boldsymbol{y})]$ and $\mathcal{L}(p_\theta(\boldsymbol{x})) \triangleq$

$\mathbb{E}_{p_t(\boldsymbol{x})}[-\log p_{\boldsymbol{\theta}}(\boldsymbol{x})]$, we obtain from the latent variable model setup in Section 2 that

$$\mathcal{L}(p_{\boldsymbol{\theta}}(\boldsymbol{x})) \leq \mathbb{E}_{p_t(\boldsymbol{x})}[D_{\mathrm{KL}}(q_{\boldsymbol{\phi}}(\boldsymbol{z}|\boldsymbol{x})q_{\boldsymbol{\phi}}(\boldsymbol{s}|\boldsymbol{x}) \parallel p_t(\boldsymbol{z})p_t(\boldsymbol{s})) - \mathbb{E}_{q_{\boldsymbol{\phi}}(\boldsymbol{z}|\boldsymbol{x})q_{\boldsymbol{\phi}}(\boldsymbol{s}|\boldsymbol{x})} \log p_{\boldsymbol{\theta}}(\boldsymbol{x}|\boldsymbol{z},\boldsymbol{s})],$$

and

$$\begin{aligned}
\mathcal{L}(p_{\boldsymbol{\theta}}(\boldsymbol{y})) &\leq \mathbb{E}_{p(\boldsymbol{y})}\left[ D_{\mathrm{KL}}(q_{\boldsymbol{\phi}}(\boldsymbol{z}|\boldsymbol{y})q_{\boldsymbol{\phi}}(\boldsymbol{s}|\boldsymbol{y}) \parallel p_b(\boldsymbol{z})p_b(\boldsymbol{s})) - \mathbb{E}_{q_{\boldsymbol{\phi}}(\boldsymbol{z}|\boldsymbol{y})q_{\boldsymbol{\phi}}(\boldsymbol{s}|\boldsymbol{y})} \log p_{\boldsymbol{\theta}}(\boldsymbol{y}|\boldsymbol{z},\boldsymbol{0}) \right] \\
&= \mathbb{E}_{p(\boldsymbol{y})}[D_{\mathrm{KL}}(q_{\boldsymbol{\phi}}(\boldsymbol{s}|\boldsymbol{y}) \parallel \delta\{\boldsymbol{s}=\boldsymbol{0}\})] \\
&\qquad\qquad + \mathbb{E}_{p(\boldsymbol{y})}\left[ D_{\mathrm{KL}}(q_{\boldsymbol{\phi}}(\boldsymbol{z}|\boldsymbol{y}) \parallel p_b(\boldsymbol{z})) - \mathbb{E}_{q_{\boldsymbol{\phi}}(\boldsymbol{z}|\boldsymbol{y})} \log p_{\boldsymbol{\theta}}(\boldsymbol{y}|\boldsymbol{z},\boldsymbol{0}) \right] \\
&= \mathbb{E}_{p_b(\boldsymbol{y})}[\mathbb{1}_{(\boldsymbol{0},\boldsymbol{0})}((\boldsymbol{\mu}_{\boldsymbol{\phi},\boldsymbol{s}}(\boldsymbol{y}),\boldsymbol{\sigma}_{\boldsymbol{\phi},\boldsymbol{s}}(\boldsymbol{y})))] \\
&\qquad\qquad + \mathbb{E}_{p(\boldsymbol{y})}\left[ D_{\mathrm{KL}}(q_{\boldsymbol{\phi}}(\boldsymbol{z}|\boldsymbol{y}) \parallel p_b(\boldsymbol{z})) - \mathbb{E}_{q_{\boldsymbol{\phi}}(\boldsymbol{z}|\boldsymbol{y})} \log p_{\boldsymbol{\theta}}(\boldsymbol{y}|\boldsymbol{z},\boldsymbol{0}) \right],
\end{aligned}$$

where the last equality follows from Lemma 1, and $\mathbb{1}_{\boldsymbol{c}}(\boldsymbol{x}) = 0$ if $\boldsymbol{x} = \boldsymbol{c}$ and $\infty$ otherwise, for some fixed constant $\boldsymbol{c}$, and $(\boldsymbol{\mu}_{\boldsymbol{\phi},\boldsymbol{s}}(\boldsymbol{y}), \boldsymbol{\sigma}_{\boldsymbol{\phi},\boldsymbol{s}}(\boldsymbol{y}))$ denotes the mean-variance output pair of the encoder for salient feature $\boldsymbol{s}$. Thus adding the bounds for both the target and background distributions, we obtain

$$\mathcal{L}(p_{\boldsymbol{\theta}}(\boldsymbol{x})) + \mathcal{L}(p_{\boldsymbol{\theta}}(\boldsymbol{y})) \leq \mathbb{E}_{p_b(\boldsymbol{y})}[\mathbb{1}_{\{(\boldsymbol{0},\boldsymbol{0})\}}((\boldsymbol{\mu}_{\boldsymbol{\phi}}(\boldsymbol{y}),\boldsymbol{\mu}_{\boldsymbol{\sigma}}(\boldsymbol{y})))] + L_{\mathrm{VAE}} \triangleq L_{\mathrm{c}} + L_{\mathrm{VAE}}. \tag{6}$$

We now define the upper bound in Eq. (6) as our new VAE loss, i.e.,

$$L_{\mathrm{VAE}}^{(\mathrm{ours})} \triangleq \mathbb{E}_{p_b(\boldsymbol{y})}[\mathbb{1}_{\{(\boldsymbol{0},\boldsymbol{0})\}}((\boldsymbol{\mu}_{\boldsymbol{\phi},\boldsymbol{s}}(\boldsymbol{y}),\boldsymbol{\sigma}_{\boldsymbol{\phi},\boldsymbol{s}}(\boldsymbol{y})))] + L_{\mathrm{VAE}}. \tag{7}$$

Note that our indicator term above strictly enforces the salient feature $\boldsymbol{s}$ to be zero for the background dataset. This directly reflects the problem structure of contrastive disentanglement. In particular, this constrains the encoder to encode the background information in the common factor $\boldsymbol{z}$ alone, encouraging contrastive disentanglement. We empirically validate this claim in Section 4.1 and show that our new loss $L_{\mathrm{VAE}}^{(\mathrm{ours})}$ outperforms the existing approaches on a variety of datasets and benchmarks. Since the indicator loss is non-differentiable, for the sake of implementational ease, we approximate it by a quadratic loss $\mathbb{E}[\|\boldsymbol{\mu}_{\boldsymbol{\phi},\boldsymbol{s}}(\boldsymbol{y})\|^2 + \|\boldsymbol{\sigma}_{\boldsymbol{\phi},\boldsymbol{s}}(\boldsymbol{y})\|^2]$. In fact, this loss term is motivated by the following fact (which follows from Lemma 2) that

$$W_2^2(q_{\boldsymbol{\phi}}(\boldsymbol{s}|\boldsymbol{y}), \delta\{\boldsymbol{s}=\boldsymbol{0}\}) = \|\boldsymbol{\mu}_{\boldsymbol{\phi},\boldsymbol{s}}(\boldsymbol{y})\|^2 + \|\boldsymbol{\sigma}_{\boldsymbol{\phi},\boldsymbol{s}}(\boldsymbol{y})\|^2,$$

where $W_2^2(\cdot,\cdot)$ denotes the second-order Wasserstein squared distance. Hence in view of the above equation, our Wasserstein distance based quadratic regularizer acts a differentiable surrogate to the non-differentiable KL divergence term (which is exactly the indicator term) in Eq. (7) above.

## 3.2 WASSERSTEIN LOSS FOR DISTRIBUTIONAL SIMILARITY

In the above section, we discussed why the new loss $L_{\mathrm{VAE}}^{(\mathrm{ours})}$ is able to achieve better disentangled representations than current state-of-the-art approaches. In addition to obtaining a disentangled set of latent features $(\boldsymbol{z}, \boldsymbol{s})$, an equally important and additional core goal of contrastive disentanglement is that the distribution of the common factor $\boldsymbol{z}$ remain identical across both datasets. However, experiments in Section 4.2 indicate that our proposed $L_{\mathrm{VAE}}^{(\mathrm{ours})}$ loss as well as losses of existing frameworks do not meet this criterion. While the assumption of the same latent prior for the common factor across both datasets, i.e., $p_b(\boldsymbol{z}) = p_t(\boldsymbol{z})$, and sharing of the encoder architecture for both datasets implicitly enforce this desired requirement, empirical evidence suggests that current losses do not strictly enforce it. This motivates to define a new loss $L_{\mathrm{W}_2,\mathrm{VAE}}^{(\mathrm{ours})}$ that addresses this distributional mismatch via

$$L_{\mathrm{W}_2,\mathrm{VAE}}^{(\mathrm{ours})} \triangleq L_{\mathrm{VAE}}^{(\mathrm{ours})} + \lambda_{\boldsymbol{z}} \cdot W_2^2(\widehat{q}_{\boldsymbol{\phi},t}(\boldsymbol{z}), \widehat{q}_{\boldsymbol{\phi},b}(\boldsymbol{z})), \quad \lambda_{\boldsymbol{z}} > 0, \tag{8}$$

where $\widehat{q}_{\boldsymbol{\phi},t}(\boldsymbol{z}) = (1/B)\sum_{i=1}^{B} \delta\{\boldsymbol{z} = \boldsymbol{z}_i^{(t)}\}$ denotes the empirical marginal distribution of the common factors $\boldsymbol{z}_i^{(t)}$ for the target data, $\widehat{q}_{\boldsymbol{\phi},b}(\boldsymbol{z})$ is defined similarly for the background data, $B > 0$ is the batch size, and $\lambda_{\boldsymbol{z}} > 0$ is a hyperparameter. Here $W_2^2(\cdot,\cdot)$ denotes the second-order squared Wasserstein distance, defined as

$$L_{W_2}(\boldsymbol{\phi}) \triangleq W_2^2(\widehat{q}_{\boldsymbol{\phi},t}(\boldsymbol{z}), \widehat{q}_{\boldsymbol{\phi},b}(\boldsymbol{z})) = \inf_{T \in \Pi([B],[B])} \sum_{i,j} T_{ij}C_{ij}, \quad C_{ij} = \|\boldsymbol{z}_i^{(t)} - \boldsymbol{z}_j^{(b)}\|_2^2, \tag{9}$$

where $[B] \triangleq \{1,\ldots,B\}$, $\Pi([B],[B])$ denotes all joint probability distributions on $[B] \times [B]$ whose first and second marginals are both uniform distributions on $[B]$, i.e., $\Pi([B],[B]) = \{T \in \mathbb{R}_+^{B \times B} : T\mathbf{1}_B = (1/B)\mathbf{1}_B, T^\top\mathbf{1}_B = (1/B)\mathbf{1}_B\}$, and $\mathbf{1}_B$ is the vector of all ones with size $B$.

| Dataset | Method | Target Clustering | | | | $L_{W_2}$ |
|---------|--------|-------------------|---|---|---|-----------|
| | | $z_x$ | | $s_x$ | | (Lower |
| | | (Lower is better) | | (Higher is better) | | is |
| | | CA(%) | SS | CA(%) | SS | better) |
| L MNIST | cVAE | 12.59 | 0.18 | 89.02 | 0.46 | 8.91 |
| | sRb-VAE | 64.94 | 0.40 | 65.14 | 0.43 | 12.34 |
| | Ours (Eq. (7)) | 12.43 | 0.19 | 91.23 | 0.52 | 8.88 |
| | Ours (Eq. (8)) | **11.91** | **0.04** | **92.17** | **0.55** | **4.60** |
| NL MNIST | cVAE | 26.61 | 0.27 | 26.89 | **0.28** | 9.04 |
| | sRb-VAE | 22.60 | 0.16 | 23.42 | 0.16 | 38.12 |
| | Ours (Eq. (7)) | **16.36** | **0.13** | 29.44 | 0.12 | 9.02 |
| | Ours (Eq. (8)) | 17.56 | 0.19 | **32.37** | 0.16 | **4.69** |
| CelebA | cVAE | 52.06 | 0.35 | 60.01 | 0.39 | 38.26 |
| | sRb-VAE | 55.12 | 0.34 | 56.17 | 0.36 | 43.87 |
| | Ours (Eq. (7)) | **51.53** | **0.31** | 59.97 | 0.38 | 35.51 |
| | Ours (Eq. (8)) | 55.05 | 0.34 | **64.55** | **0.42** | **11.07** |
| Affectnet | cVAE | 19.02 | 0.17 | 17.75 | 0.15 | 35.76 |
| | sRb-VAE | 18.31 | 0.15 | 19.85 | 0.15 | 41.34 |
| | Ours (Eq. (7)) | 18.16 | **0.14** | 19.06 | 0.16 | 30.04 |
| | Ours (Eq. (8)) | **16.06** | **0.14** | **22.66** | **0.18** | **10.11** |

Table 1: Comparison of our methods to baselines cVAE (Eq. (4)) and sRb-VAE (Eq. (5)) evaluated on metrics CA (clustering accuracy (%)) and SS (silhouette score). Lower CA and SS on $z$ are better, meaning that the common latent vectors are indistinguishable. Ideal CA using $z$ are $10\%$ on L MNIST and NL MNIST, $50\%$ on CelebA, and $11.11\%$ on Affectnet. Lower $L_{W_2}$ is better, which means that $z_x$ and $z_y$ are distributionally similar.

Intuitively, this Wasserstein loss term $L_{W_2}$ gauges similarity of the common factors of both datasets and penalizes any distributional mismatch. This loss also ensures that the relevant target signal is encoded solely in the salient latent feature $s$. Indeed, in Section 4.2 we empirically verify this claim and establish that across a variety of datasets our new loss $L_{W_2, \text{VAE}}^{(\text{ours})}$ outpeforms existing approaches.

This Wasserstein loss has two major advantages over the KL divergence loss terms used by previous works: (1) both the loss as well as its gradients can be efficiently computed; and (2) this new loss does not require any additional trainable parameters whereas estimation of the KL divergence often requires a discriminator net, since its direct estimation from samples is hard to compute (Kim & Mnih, 2018). To compute the gradients for $L_{W_2}$, since the objective is linear in Eq. (9), it follows from Danskin's theorem (Danskin, 2012) that $\nabla_\phi L_{W_2} = \sum_{i,j} T_{ij}^* \nabla_\phi C_{ij}$, where $T^*$ is the optimal probability matrix in Eq. (9). We compute the optimal matrix $T^*$ by using the Python Optimal Transport library (Flamary & Courty, 2017) which uses an efficient linear programming solver from Bonneel et al. (2011).

## 4 EXPERIMENTS

In this section, we empirically show that our novel losses $L_{\text{VAE}}^{(\text{ours})}$ and $L_{W_2, \text{VAE}}^{(\text{ours})}$, defined in Eq. (7) and Eq. (8) respectively, perform better than the current state-of-the-art methods on a variety of datasets as highlighted in Table 1. In particular, in Section 4.1, we validate through a set of qualitative and quantitative benchmarks that the correct VAE loss $L_{\text{VAE}}^{(\text{ours})}$ gives rise to a pair of disentangled latent features as opposed to the existing approaches (also highlighted in Figure 5). Further, in Section 4.2, we empirically establish that our $L_{W_2, \text{VAE}}^{(\text{ours})}$ loss constrains the target and background common factors to have identical distributions, while the previous losses do not exhibit similar behavior. For the sake of brevity, in this section, we refer to $L_{\text{VAE}}^{(\text{ours})}$ as 'Ours (Eq. (7))' and $L_{W_2, \text{VAE}}^{(\text{ours})}$ as 'Ours (Eq. (8))'.

**Datasets**

**Linear and NonLinear Grassy MNIST.** We construct 2 synthetic datasets with signal and background images linearly and non-linearly combined respectively. The Linear Grassy MNIST dataset (L-MNIST) is constructed following Abid & Zou (2019). The background category $\{y^{(i)}\}$ consists

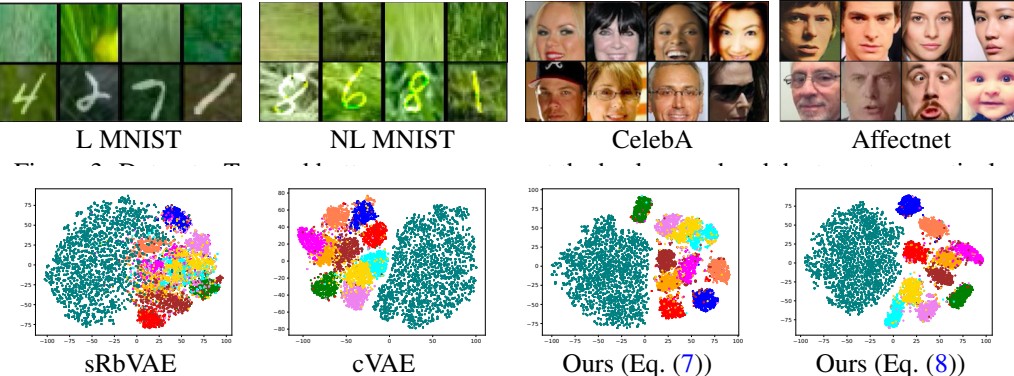

|  |  |  |  |
|---|---|---|---|
| L MNIST | NL MNIST | CelebA | Affectnet |
| sRbVAE | cVAE | Ours (Eq. (7)) | Ours (Eq. (8)) |

Figure 4: Unsupervised recovery of salient features on the Linear Grassy MNIST dataset. Teal circles represent $s$ component of grass. Other smaller clusters with multiple colors represent digits $0$ to $9$.

of randomly cropped $28 \times 28$ sized image patches from the grass category of ImageNet (Deng et al., 2009). To construct the target set $\{\boldsymbol{x}^{(i)}\}$ we choose images of digits from $0$ to $9$ from the MNIST dataset (LeCun et al., 1998) and linearly combine them with random grass images from the background. To construct the non-linear Grassy MNIST dataset (NL- MNIST), we use a thresholding operation before superimposing the MNIST digits with background patches $\boldsymbol{y}^{(i)}$, i.e., $\boldsymbol{x}^{(i)}[u, v, c] = \mathcal{I}[\boldsymbol{y}^{(i)}[u, v, c] > 0.3](\rho \boldsymbol{y}^{(i)}[u, v, c] + (1 - \rho)\boldsymbol{t}^{(i)}[u, v])$, where $\rho = 0.5$, $\boldsymbol{t}^{(i)}$ denotes the $i^{\text{th}}$ pixel of the MNIST digit image $\boldsymbol{t}$ , and $\mathcal{I} = 1$ if $\boldsymbol{y}^{(i)}[u, v, c] > 0.3$ and $0$ otherwise. Here $u$ and $v$ denote the respective row and column pixel indices of the image and $c$ is the channel index. We observe NL-MNIST to be a very challenging dataset on which the state of the art methods perform poorly. Clean class labels are available for evaluation, unlike other non-linear and challenging datasets like the CelebA (Liu et al., 2018) and Affectnet (Ruiz et al., 2019) data, which can often have noisy labels. Besides, we can control the non-linearity of this dataset by changing the threshold value. Even though the target samples $\{\boldsymbol{x}^{(i)}\}$ are generated using samples from $\boldsymbol{y}^{(i)}$, we do not have access to these pairs $(\boldsymbol{x}^{(i)}, \boldsymbol{y}^{(i)})$ during training. In Figure 3 we show samples from both datasets.

**CelebA.** We create the target and the background sets using the CelebA dataset (Liu et al., 2018) as described by Abid & Zou (2019). To form the target dataset, we use a subset of the CelebA (Liu et al., 2018), namely, faces with caps and eyeglasses. The remaining categories are shuffled and samples are randomly picked as background images.

**AffectNet.** We use this challenging dataset for disentanglement of facial expressions. To form the background, we use faces with neutral expressions as described by Ruiz et al. (2019). The target set is constructed with faces expressing some emotions, namely, happiness, sadness, surprise, fear, disgust, anger and contempt. These expressions become the salient features.

**Evaluation Metrics**

**Clustering Accuracy and Silhouette Score.** To empirically evaluate that the learned salient features $\boldsymbol{s}_x$ of the target dataset correspond to the ground truth class labels, we use the *silhouette score* (SS) and *clustering accuracy* as our metrics. For both the metrics, first we perform t-SNE (Maaten & Hinton, 2008) on the target salient features to embed them in a two dimensional latent space and cluster them using the standard k-means. The clustering accuracy computes the proportion of the samples whose class labels align with the ground-truth labels. The SS metric gauges the similarity between the clustering labels and the ground-truth class labels by comparing the intra-cluster distances to inter-cluster distances. The SS score takes values between $-1$ to $1$. The higher the score, the stronger the correspondence between the cluster labels and the class labels. Furthermore, to verify that the common factors $\boldsymbol{z}_x$ of the target dataset don't encode salient features, we cluster the common factors using the same procedure as above and empirically validate whether the clustering accuracy and the silhouette score are close to that of a random label assignment. For example, for Grassy MNIST, allotment of random labels (MNIST digit values) would achieve $10\%$ accuracy.

**Wasserstein loss between common factors.** We use the Wasserstein loss $L_{W_2}$ defined in Eq. (9) to measure the similarity between the empirical distributions of the target common factors and the background common factors respectively. The closer it is to zero, the larger the similarity.

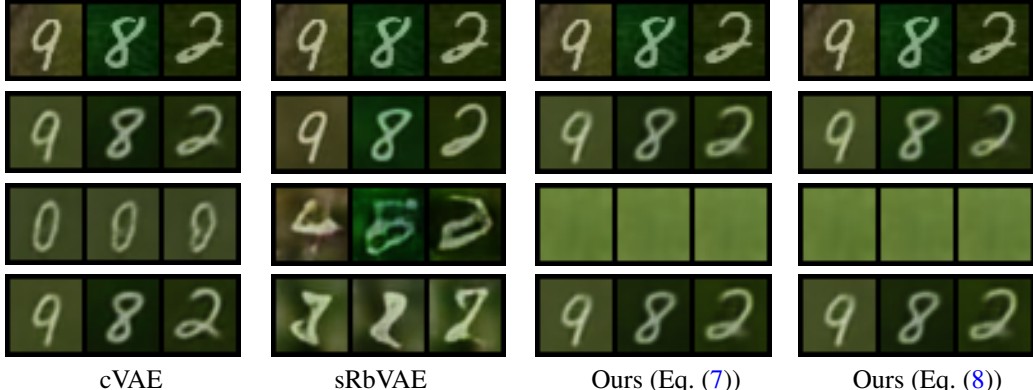

cVAE          sRbVAE          Ours (Eq. (7))          Ours (Eq. (8))

Figure 5: Disentanglement of different methods. Rows from top to bottom represent target images $x$ from the L-MNIST dataset, decoded images from passing $(z, s)$, $(z, 0)$, and $(0, s)$ through the decoder respectively. Note that the latent vector $(z, s)$ is obtained after the forward pass of the target image $x$ through the encoder.

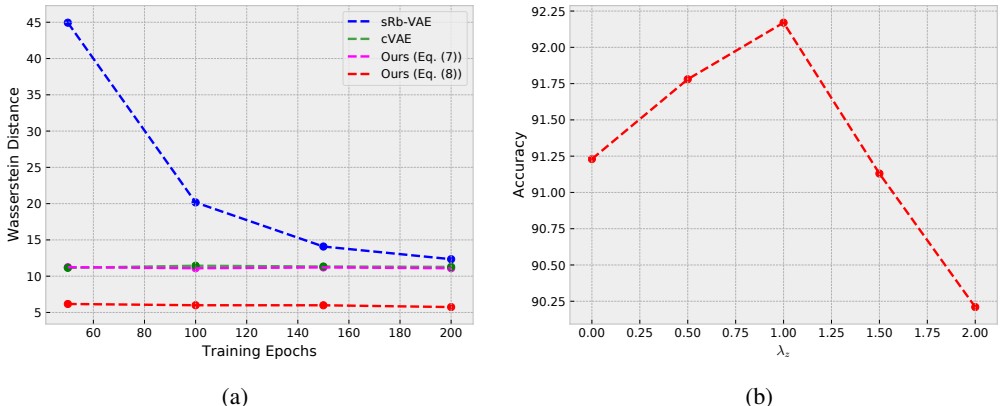

(a)                                    (b)

Figure 6: (a) $L_{W_2}$ loss across training. Ours (Eq. (8)) optimizes over $L_{W_2}$ (red plot) and hence has a very low $L_{W_2}$ value after 200 epochs. (b) Clustering accuracies over $\lambda_z$ influencing $L_{W_2}$ for Linear MNIST. $\lambda_z = 1$ achieves the best value across multiple trials.

**Affectnet metric: Classification.** In addition, we also use classification accuracy to gauge the correspondence between the salient latent feature and the ground-truth class label. In particular, we train a shallow classifier with the latent feature and the respective class label as the input-output training samples and report the classification accuracy of this trained model.

## 4.1 CORRECT VAE LOSS

This section empirically demonstrates that our correct VAE loss $L_{\text{VAE}}^{(\text{ours})}$ significantly improves disentanglement over that of cVAE and sRb-VAE. Quantitatively, in Table 1, we conduct a thorough comparison of our approach to prior work across all the aforementioned datasets. We observe that the correct VAE loss is able to consistently obtain better disentangled features as reflected in the clustering accuracies of both the salient features $s_x$ and the common features $z_x$. Qualitatively, in Figure 4, for the L-MNIST dataset, the salient features of the target and background samples, i.e., $s_x$ and $s_y$, are embedded in a two dimensional latent space using tSNE and then clustered using k-means. Note that $s_y$ is expected to be close to zero, whereas $s_x$ is supposed to encode the information about the MNIST digits. As shown in Figure 4, clusters obtained through our loss are well separated compared to that of both cVAE and sRb-VAE. Indeed, we achieve a clustering accuracy of $91\%$ on the L-MNIST dataset over all the 10 digit classes, while cVAE and sRb-VAE attain accuracies of $89\%$ and $65\%$ respectively. Figure 5 further illustrates that the target common factors $z_x$ and the salient factors $s_x$, obtained with cVAE and sRb-VAE, contain information about the salient features and the background features respectively. This is contrary to the goal of contrastive disentanglement. Instead, our reconstructed samples show no leakage of information.

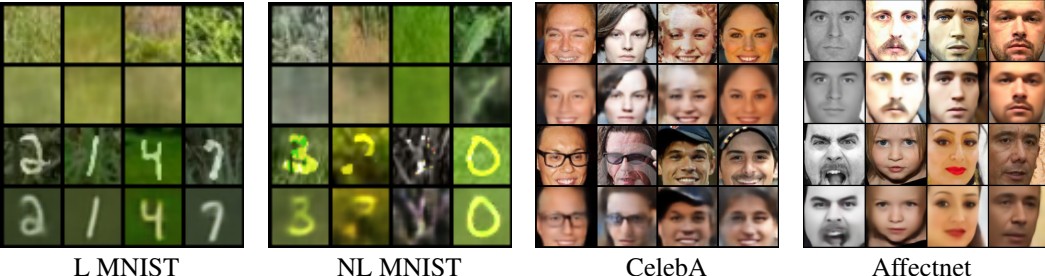

| L MNIST | NL MNIST | CelebA | Affectnet |

Figure 7: Reconstruction results with Ours (Eq. (8)). The rows from top represent the background images from the dataset, reconstructed background, target images from the dataset, and reconstructed target images respectively.

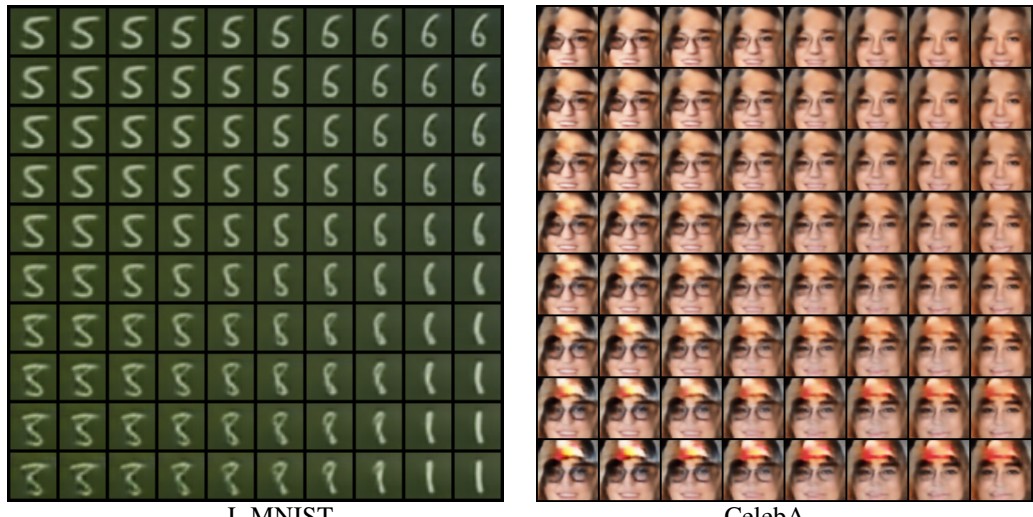

| L MNIST | CelebA |

Figure 8: Interpolation results. Interpolating from $s_1$ to $s_2$ along the diagonal, keeping $z_1$ fixed. Top left: $[z_1, s_1]$ decoded. Bottom Right $[z_1, s_2]$ decoded.

## 4.2  $L_{W_2}$ LOSS

In the earlier section, we empirically demonstrated that the correct VAE loss is able to obtain better disentangled representations than the baselines. However, as highlighted in Figure 6, all these losses do not enforce the distributions of the common factors across the target and background datasets to be identical. This is illustrated by a high Wasserstein loss (also shown in Table 1). Instead, in the presence of our new loss $L_{W_2,VAE}^{(ours)}$, we observe that the corresponding loss values reduce considerably. This is highlighted in Figure 6 for the L-MNIST dataset and in Table 1 across all the datasets. In addition, we also observe in Table 1 that encouraging the distribution of the common factors to be identical across both datasets via $L_{W_2,VAE}^{(ours)}$ further enhances the disentanglement between the common and the salient features. This is shown via the higher clustering accuracies compared to baselines. Intuitively, enforcing the common factors to have the same distribution implicitly constrains the unique features of the target dataset to be encoded in the salient features alone. This phenomenon is also illustrated in Figure 5. We found the hyperparameter choice $\lambda_z = 1$ to perform best across all datasets. Figure 6 (right) shows the clustering accuracies of digits for L-MNIST as we vary $\lambda_z$.

Figure 7 shows some reconstruction results of the background and target images on different datasets, obtained from the approach described in this section. Figure 8 shows the results obtained by interpolating 2 classes of the target salient factors in 2 dimensions. This is performed on the L MNIST and CelebA datasets.

## 5  RELATED WORK

Unsupervised contrastive analysis has been proposed very recently in work by Abid et al. (2017) and Severson et al. (2018). Inspired by the principal component analysis (PCA), Abid et al. (2017) describe 'contrastive PCA' (cPCA), a model which discovers low-dimensional structure that uniquely characterizes a dataset compared to another. The approach linearly transforms latent representations

| | Happiness | Sadness | Surprise | Fear | Disgust | Anger | Contempt |
|---|---|---|---|---|---|---|---|
| DIP-VAE-II | 0.548 | 0.245 | **0.401** | **0.389** | 0.268 | 0.391 | 0.463 |
| sVAE | 0.583 | 0.251 | 0.389 | 0.349 | 0.260 | 0.391 | 0.469 |
| $\beta$-TCVAE | 0.563 | 0.277 | 0.393 | 0.349 | 0.256 | **0.427** | 0.467 |
| RbVAE | 0.536 | 0.393 | 0.379 | 0.311 | 0.320 | 0.383 | 0.421 |
| sRbVAE | **0.587** | 0.405 | 0.387 | 0.327 | 0.344 | 0.425 | 0.483 |
| Ours(Eq. (7)) | 0.575 | 0.488 | 0.389 | 0.365 | 0.356 | 0.395 | 0.501 |
| Ours(Eq. (8)) | 0.579 | **0.491** | **0.401** | 0.374 | **0.358** | 0.398 | **0.506** |

Table 2: Affectnet (Ruiz et al., 2019). Per class classification accuracy on each class of emotions.

which are also linearly combined. To obtain more expressive transformations kernels have also been investigated (Abid et al., 2017). Since linearity restricts expressiveness of the model, in more recent work, Severson et al. (2018) introduce 'contrastive latent variable models' which permit to non-linearly transform the latent representation. Importantly, contrastive latent variable models still combine transformations linearly.

Disentangling of representations (Bengio et al., 2013) has also been investigated in the computer vision community. Recently, a variety of approaches like $\beta$-VAE (Higgins et al., 2016), DIP-VAE (Kumar et al., 2018), FactorVAE (Kim & Mnih, 2018) or $\beta$-TCVAE (Chen et al., 2018) have been discussed. Generally those techniques operate on a single dataset with the goal to extract latent factors.

Also related is work on reference-based variational auto-encoders (Rb-VAEs) (Ruiz et al., 2018), where reference-based disentangling is introduced. A model is carefully analyzed and the resulting cost function is, augmented via a reverse KL-distribution to disentangle factors between two datasets.

In contrast, to the aforementioned methods we develop a model that disentangles factors without the need for any heuristics. To this end we introduce two losses. The first loss encourages the encoded background data representation to avoid any signal in the salient feature. The second loss encourages the non-salient probability distributions to be identical for both the background and the target set. Both losses combined help us accurately disentangle salient representations.

Canonical correlation analysis (CCA) (Hotelling, 1936) and its probabilistic counterpart (PCCA) (Bach & Jordan, 2005) also operate on at least two sets of data. However, both require the datasets to be paired. This is also true for the non-linear extensions, e.g., the one based on Gaussian processes (Damianou et al., 2016).

Dimensionality reductions techniques like t-SNE (Maaten & Hinton, 2008) and multi-dimensional scaling (MDS) (Cox & Cox, 2000) are related in that they recover non-linear data projections. Yet, they are designed to explore one dataset at a time. Consequently, for a contrastive analysis, those approaches are applied on each dataset separately and a manual comparison is subsequently required to uncover similarities and differences.

Statistical test, e.g., two-sample t-test, Fisher's discriminant analysis, Wilcoxon signed-rank test, Mann-Whitney U-test, identify differences between two datasets given features. Albeit uncovering feature differences, none of those approaches are developed to find differentiating features.

## 6 CONCLUSION

In this paper we present two principled losses for variational autoencoder based models which address contrastive disentanglement, i.e., extracting the salient features that enhance one dataset (target) compared to another (background). Our first loss explicitly discourages expression of salient features for the background data and is derived from classical variational principles. The second loss encourages background features to be identical for both target and background data. In extensive experiments on a variety of datasets we showed the benefits of the two introduced losses.

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

# A    APPENDIX

**Lemma 1.** *Let $P = \mathcal{N}(s; \boldsymbol{\mu}, \boldsymbol{\sigma}^2)$ be a Gaussian distribution with mean $\boldsymbol{\mu}$ and diagonal covariance $\boldsymbol{\sigma}^2$, and $Q = \delta\{s = 0\}$ be a Dirac distribution centred at zero. Then*

$$D_{\mathrm{KL}}(P \parallel Q) = \mathbb{1}_{(\mathbf{0},\mathbf{0})}((\boldsymbol{\mu}, \boldsymbol{\sigma})),$$

*where $\mathbb{1}_{(\mathbf{0},\mathbf{0})}((\boldsymbol{\mu}, \boldsymbol{\sigma})) = 0$, if $(\boldsymbol{\mu}, \boldsymbol{\sigma}) = (\mathbf{0}, \mathbf{0})$, and $\infty$ otherwise.*

*Proof.* By definition, we know that $D_{\mathrm{KL}}(P \parallel Q) = \mathbb{E}_P[\log \frac{P(s)}{Q(s)}]$ if $P \ll Q$ and $\infty$ otherwise. Since $P$ is a Gaussian and $Q$ is a Dirac, we have that $P \ll Q$ if and only if $(\boldsymbol{\mu}, \boldsymbol{\sigma}) = (\mathbf{0}, \mathbf{0})$. Hence the claim follows. □

**Lemma 2.** *Let $P = \mathcal{N}(s; \boldsymbol{\mu}, \boldsymbol{\sigma}^2)$ be a Gaussian distribution with mean $\boldsymbol{\mu}$ and diagonal covariance $\boldsymbol{\sigma}^2$, and $Q = \delta\{s = 0\}$ be a Dirac distribution centred at zero. Then*

$$W_2^2(P, Q) = \|\boldsymbol{\mu}\|^2 + \|\boldsymbol{\sigma}\|^2.$$

*Proof.* Note that $W_2^2(P, Q)$ is defined as

$$W_2^2(P, Q) = \inf_{P_{X,Y}: P_X = P, P_Y = Q} \mathbb{E}\|X - Y\|^2,$$

where the infimum is over all joint probability distributions $P_{X,Y}$ such that $X$ has the marginal $P$, whereas $Y$ follows $Q$. Since $P$ is a Gaussian distribution, which is absolutely continuous with respect to Lebesgue measure in $\mathbb{R}^d$, it follows from Theorem 2.12 of Villani (2003) that

$$W_2^2(P, Q) = \inf_{T: T\#P = Q} \mathbb{E}_{X \sim P}\|X - T(X)\|^2,$$

where $T\#P = Q$ denotes that the pushforward of probability measure $P$ under the map $T : \mathbb{R}^d \to \mathbb{R}^d$ is $Q$. Or equivalently, $T(X) \sim Q$ whenever $X \sim P$. Since $Q$ is a Dirac mass at zero, it follows that the only such feasible transport map is given by $T(x) = 0, \forall x \in \mathbb{R}^d$. Hence, $W_2^2(P, Q) = \mathbb{E}_{X \sim P}\|X\|^2 = \|\boldsymbol{\mu}\|^2 + \|\boldsymbol{\sigma}\|^2$. □

## A.1    DATASET DESCRIPTION

*Linear and NonLinear Grassy MNIST*

For the background dataset $\{y^{(j)}\}$, we randomly crop $28 \times 28$ sized image patches from the grass category of the ImageNet dataset (Deng et al., 2009). We use a total of 4986 grass patches to construct this dataset. We construct the target dataset $\{x^{(i)}\}$ in the following way: We choose 4986 samples for each digit from 0 to 9 using the MNIST dataset. We use 4986 samples because it corresponds to the lowest number of occurrences for a digit (digit 4). Each of these digits are then superimposed on random grass images from the background dataset. The superimposition is a linear operation described by

$$x^{(i)} = \rho y^{(i)} + (1 - \rho)t^{(i)}, \tag{10}$$

where $t^{(i)}$ is a random MNIST image and $\rho = 0.5$. Even though the samples $\{x^{(i)}\}$ are generated from $y^{(i)}$ according to Eq. (10), we do not have access to these pairs $(x^{(i)}, y^{(i)})$ during training.

To construct the non-linear dataset, we use a thresholding operation before superimposing the MNIST digits on $y^{(i)}$, i.e.

$$x^{(i)}[u, v, c] = \mathcal{I}[y^{(i)}[u, v, c] > 0.3](\rho y^{(i)}[u, v, c] + (1 - \rho)t^{(i)}[u, v]) \tag{11}$$

where $t^{(i)}$ is a random MNIST image, $\rho = 0.5$, $\mathcal{I}$ denotes the indicator function, $u, v$ denote the row and column pixel indices of the image and $c$ is the channel number.

*CelebA*

We form the target and the background sets using the CelebA dataset (Liu et al., 2018) as described by Abid & Zou (2019). To form the target dataset, we use a subset of the CelebA (Liu et al., 2018), namely, faces with caps and eyeglasses. For each class, we take $5000$ samples. The remaining categories are shuffled and $10,000$ samples are randomly picked as background images. Each image from the target and background is cropped into $64 \times 64$ patches.

*AffectNet*

Use this challenging dataset for disentanglement of facial expressions. Background: faces with neutral expressions. Target: Faces with other expressions, namely happy, sad, surprised, fearful, disgust, angry and contemptuous. Labels only for qualtitative evaluation. (collected from affectnet cite, adria cite) Aligned and cropped (cite adria). $96 \times 96$

## A.2 SOME RECONSTRUCTION AND INTERPOLATION RESULTS

Figure 9 shows the reconstructed results across the different datasets. From top, the rows represent background images, reconstructed background, target images and reconstructed target respectively. Figure 10 shows results by interpolating salient features of target across 2 dimensions. The common factors are kept constant

## A.3 ARCHITECTURAL CHOICES

We use a 6 layer architecture for our model to test it on the Linear and Non-Linear Grassy MNIST dataset. The encoder consists of 5 convolutional layers and 1 fully connected layer. LeakyReLU activation function is used as the non-linearity and channel normalization is used after each convolutional layers. The decoder consists of 1 fully connected and 5 transposed convolutional layers. We use the LeakyReLU activation for the initial layers and the sigmoid function for the last layer. We use a batch size of 128, a learning rate of $10^{-3}$ and Adam optimizer (Kingma & Ba, 2014). We performed all the experiments using an 16 dimensional latent space to capture the grass and a 8 dimensional space to capture digits, i.e., $z_b \in \mathbb{R}^8, z_t \in \mathbb{R}^4$. The weight on the KLD term in the VAE loss is $\beta = 2$ (Higgins et al., 2017).

We adopt a 8 layer architecture for our model to be consistent with (Abid & Zou, 2019) for evaluation on CelebA dataset. We performed all the experiments using an 16 dimensional latent space to capture the background faces and a 6 dimensional space to capture target variations, namely eyeglasses or caps.

For Affectnet, we use a 8 layer architecture for our model to be consistent with (Ruiz et al., 2019). We tale equal number of target images (5000) per class performed all the experiments using an 32 dimensional latent space to capture the background faces and a 32 dimensional space to capture target variations, namely eyeglasses or caps.

## A.4 ADDITIONAL EXPERIMENTS

We have performed additional experiments with varied sample sizes and found that our methods still consistently outperform the existing approach from Ruiz et al. (2019) across a variety of sample sizes. Figure 11 shows the results of this experiment. The experimental details are as follows: In our experiments, we found that Ruiz et al. (2019) performs the best among all the other baselines as seen from Table 1 of our paper, hence we consider Ruiz et al. (2019) to be our baseline. With regards to varied sample sizes, for the target dataset containing MNIST linearly superimposed on grass (L-MNIST), we chose the number of samples from each digit class to be 1000, 2500 and 4986. Similarly, for the background dataset of just grass images, the total number of samples are 4986, 2500 and 1000 respectively.

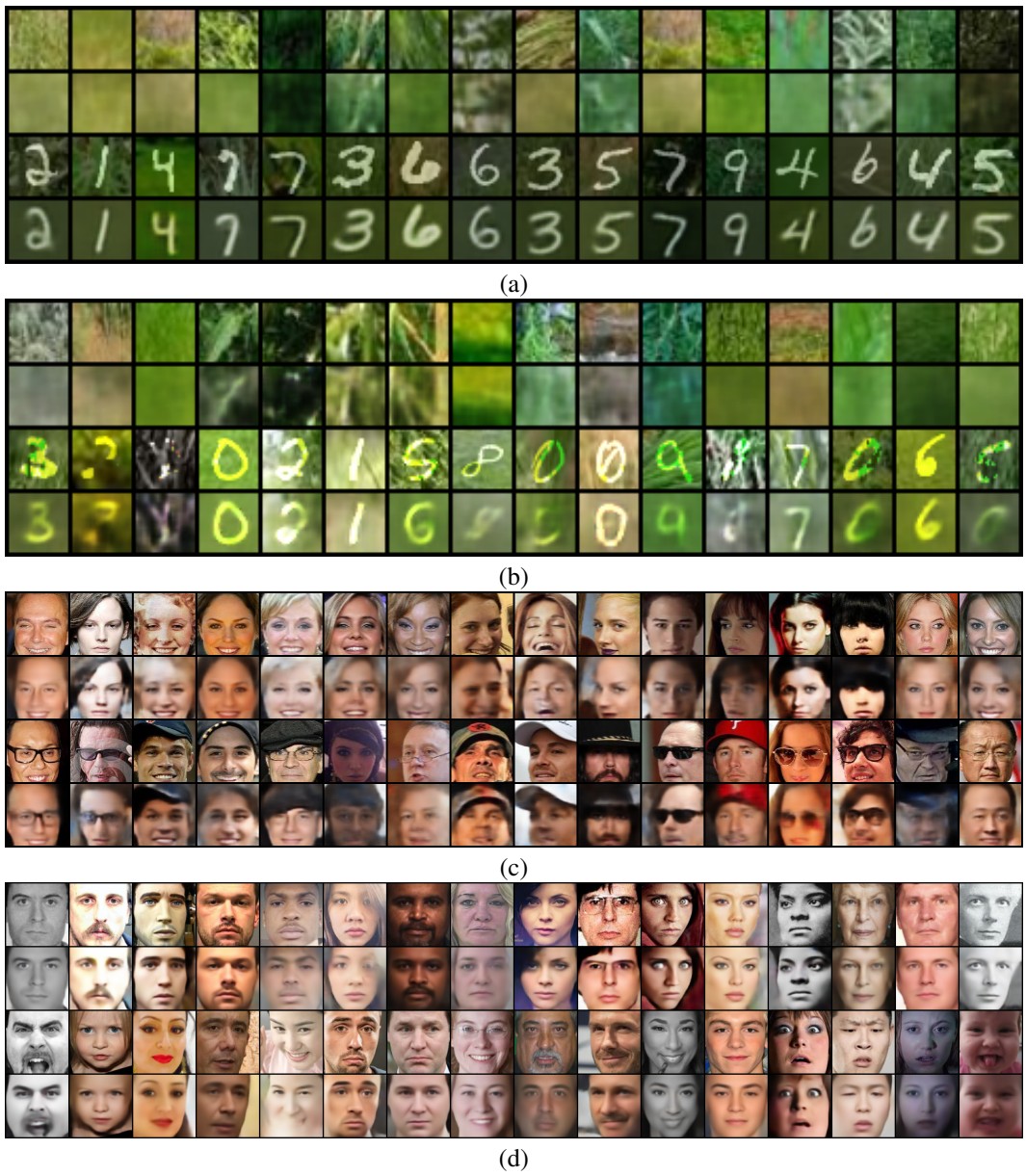

Figure 9: Reconstruction results across different datasets. (a) L MNIST, (b) NL MNIST, (c) CelebA, (d) Affectnet

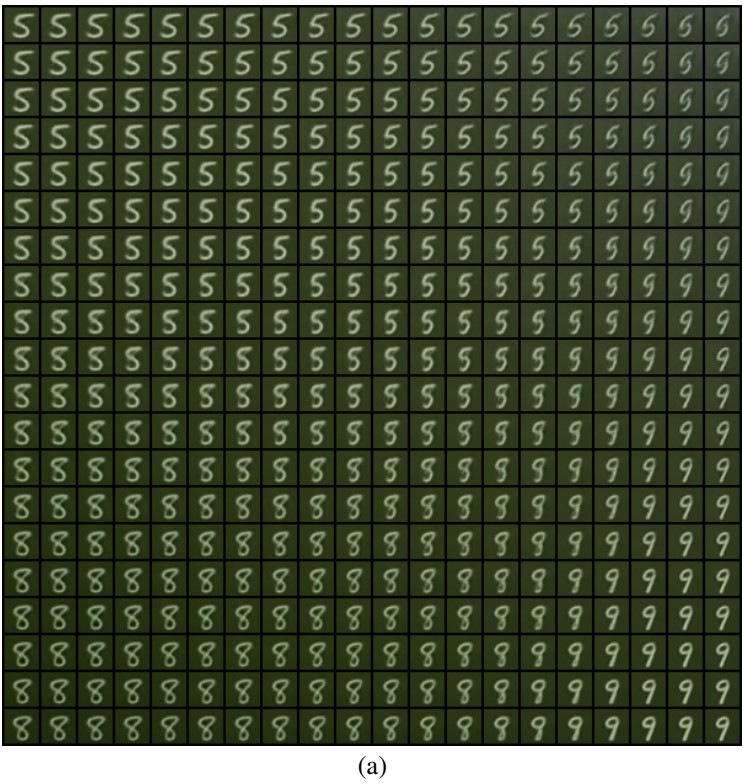

(a)

Figure 10: 2D Interpolation on L MNIST

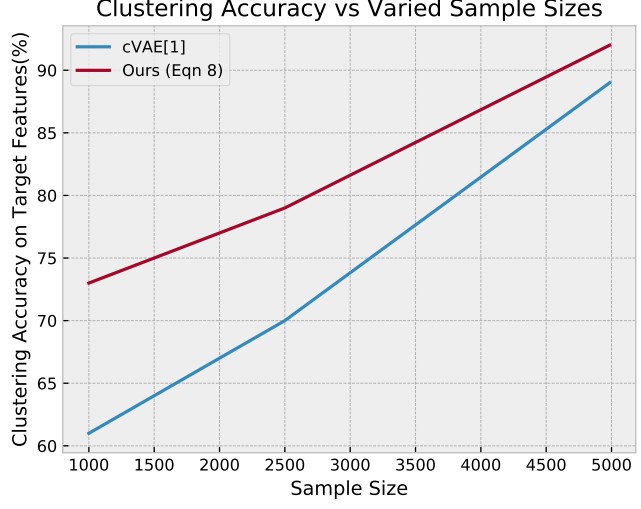

Figure 11: Clustering Accuracy(%) versus sample sizes for cVAE (Ruiz et al., 2019) and for our approach (Eqn 8)

