# OpenReview forum: "Towards Principled Objectives for Contrastive Disentanglement"
_ICLR.cc/2020/Conference — Reject_

### Official Review · AnonReviewer1 · 2019-10-25
**Official Blind Review #1**

**Rating:** 3

**Review:**

Contributions:
1. This paper fixes two problems that appeared in common contrastive disentanglement methods.
2. The paper evaluates its method in multiple experiments.

Overall, this paper does a nice contribution to improving existing methods of contrastive disentanglement. My major concern is the novelty in this paper since all the contributions can be summarized into adding two additional loss terms, which I would like to discuss below.

1. [The missing term KL[q(s|y)||p_y(s)]]. One major claim of this paper is to add back the additional term KL[q(s|y)||p_y(s)] in order to push the encoder q(s|y) to converge to a point mass \delta\{s=0\} for background images y. According to the mathematical formula, this term is natural, but the major concern is the non-overlapping support between q(s|y) and p_y(s), since in the paper they assume p_y(s)=\delta\{s=0\}. This paper claims they fix this problem. But I do not think the solution is satisfactory, because they use another term E_q[\mu_{\phi,s}(y)^2+\sigma2_{\phi,s}(y)] to replace KL[q(s|y)||\delta\{s=0\}]. If I get it correct, the replacing term E_q[\mu(y)^2+\sigma2(y)]=H(q(s|y)) is the entropy of q(s|y). But if we check the true divergence term, KL[q(s|y)||\delta\{s=0\}]=-E_q[\log\delta\{s=0\}]-H(q(s|y)) that consists of a negative entropy term. This means the behavior of the proposed loss tries to minimize the entropy while the original term tries to maximize the entropy, regardless of it is ill-defined. Thus, the replacement of the loss term seems reasonable to me at first glance but does not quite directly fix the problem of an ill-defined KL-divergence. A more reasonable solution is to replace the KL divergence with the Wasserstein distance. Is there any difficulty if we do that?

2. [The additional distributional matching term W_2^2(q_t(z),q_b(z))]. This term makes perfect sense to me, even though it is just added artificially. I'm more curious about the way to evaluate the gradient. Normally, computing Wasserstein distance is hard due to a hard linear programming matching algorithm. In this paper, the authors seem to use an existing library to compute the optimal transportation matrix. Is that a heavy computational burden? Or one could resort to some entropy regularization methods with sinkhorn iterative solvers. I'm curious about how these two methods compare with each other.

3. Another concern is the numerical result seems not as good as the qualitative results.

Overall, I think this paper does make some changes regarding previous works on contrastive disentanglement. But their main contributions need some more explanation and justification.

**Experience Assessment:**

I have read many papers in this area.

**Review Assessment: Checking Correctness Of Derivations And Theory:**

I carefully checked the derivations and theory.

**Review Assessment: Checking Correctness Of Experiments:**

I carefully checked the experiments.

**Review Assessment: Thoroughness In Paper Reading:**

I read the paper thoroughly.

---

> ### Author Response · Authors · 2019-11-14
> **Response to Review #1**
>
> We really appreciate your encouraging words on our contributions.
>
>
> “If I get it correct, the replacing term $E_q[\mu(y)^2+\sigma(y)^2]=H(q(s|y))$ is the entropy of $q(s|y)$.”
>
> We would like to clarify that our proposed loss term $E_q[\mu(y)^2+\sigma(y)^2]$ does not equal to the entropy $H(q(s|y))$. The reason: $q(s|y)$ is a Gaussian distribution with a diagonal covariance matrix $\Sigma$, its differential entropy is given by $H(q(s|y)) = \frac{1}{2} \log(det(2 \pi e\Sigma))$ [1]. Note that the diagonal entries of $\Sigma$ are denoted by the variance vector $\sigma(y)$. Moreover, the entropy $H(q(s|y))$ does not depend on the mean $\mu(y)$ because differential entropy is translation invariant. Thus our loss term is not the same as the Gaussian entropy.
>
>
> “This means the behavior of the proposed loss tries to minimize the entropy while the original term tries to maximize the entropy, regardless of it is ill-defined. Thus, the replacement of the loss term seems reasonable to me at first glance but does not quite directly fix the problem of an ill-defined KL-divergence.”
>
> As mentioned, our proposed loss does not nullify the original loss which maximizes entropy: the loss term is not the Gaussian entropy. We also like to emphasize that KL divergence between any two distributions $P$ and $Q$, i.e., $KL(P || Q)$, is always well defined [2]. If $P$ is not absolutely continuous with respect to $Q$, which is the case here since $Q$ is a Delta distribution, it is simply defined as infinity. The fact that we obtain a term containing KL divergence between a Gaussian and a Delta distribution follows naturally from the classical VAE style upper bound on the negative log-likelihood. We propose a new differentiable loss term instead of this KL divergence so as to obtain gradients.
>
>
> “ A more reasonable solution is to replace the KL divergence with the Wasserstein distance”
>
> This is a very good suggestion. Indeed, our proposed loss term $E_q[\mu(y)^2+\sigma(y)^2]$ exactly equals the squared second-order Wasserstein distance $W_2^2(q(s|y), \delta\{s=0\} )$ averaged over $y$. We apologize for not stressing upon this fact in the original version. Our updated version highlights this contribution. In particular, we have provided a mathematical proof for the above fact as Lemma 2 in the Appendix of our revised paper.
>
>
> “I'm more curious about the way to evaluate the gradient.... Is that a heavy computational burden?”
>
> To compute the Wasserstein loss $W_2^2(q_t(z),q_b(z))$, we use mini-batches of samples from both the distributions $q_t(z)$ and $q_b(z)$. The batch size is 64. Hence we can efficiently compute the optimal matching matrix using standard optimal transport libraries such as [4]. With regards to computing the gradient, if $T^\ast$ denotes the optimal transport matrix and $C$ is the cost matrix which depends on the VAE parameters $\phi$, the gradient of $W_2$ loss term is given by $\nabla_\phi L = \sum_{i,j} T^\ast_{ij} \nabla_\phi C_{ij}$. Note that this follows from Danskin’s theorem [5]. This is also highlighted in the last paragraph of Section 3.2 of the paper. Hence our proposed loss only incurs a negligible additional computational cost. In fact this is reflected by the running times of our algorithm with and without the loss $L_{W_2}$:
>
> Time to run 1 epoch:
> Ours (Eqn 7) [without $L_{W_2}$]: 16.9 secs
> Ours (Eqn 8) [with $L_{W_2}$]: 17.3 secs
>
>
> “Another concern is the numerical result seems not as good as the qualitative results.”
>
> We respectfully disagree. Please note that in Table 1, for columns denoted by $z_x$, both lower clustering accuracy (CA) and lower silhouette score (SS) indicate better performance, whereas for columns titled $s_x$, higher CA and higher SS are better. This is also explained in the caption of Table 1. To clarify this confusion we redesigned this table. Thus our method shows a healthy and consistent 3% improvement on linear MNIST, 5% improvement on non-linear MNIST, 4% improvement on CelebA and 5% improvement on Affectnet datasets. This clearly demonstrates the superiority of our method over the existing state-of-the-art approaches. We obtain better performance with much fewer number of parameters to train and using principled and well motivated loss functions. Hence we believe our qualitative and quantitative results show significant improvements over the state-of-the-art.
>
>
> References:
> [1]  https://en.wikipedia.org/wiki/Multivariate_normal_distribution#Differential_entropy
> [2] Definition 1.4, Page 18 in http://www.stat.yale.edu/~yw562/teaching/itlectures.pdf
> [3] Theorem 2.12 of Topics in Optimal Transportation, Cedric Villani. Link: https://people.math.gatech.edu/~gangbo/Cedric-Villani.pdf
> [4] Python Optimal Transport: https://github.com/rflamary/POT
> [5] https://www.jstor.org/stable/2946123

---

### Official Review · AnonReviewer3 · 2019-10-26
**Official Blind Review #3**

**Rating:** 3

**Review:**

Existing formulations for contrastive disentanglement ignore the information about the prior, i.e., the salient features of the background data should be zero, and moreover, additional KL-divergence-based losses which are hard to estimate in practice are introduced to improve disentanglement.

To resolve this issue, the authors propose new regularizations still based on VAE. Specifically, they propose to

Regularization 1. penalize the generated mean and standard deviation of background latent variable towards (0,0) which means Dirac-delta distribution on 0.

Regularization 2. match common latent variable distributions for background dataset and target dataset using Wasserstein distance.

While authors point out that existing formulation in Abid & Zou (2019); Ruiz et al. (2019) ignores about the prior, the effect of regularizing salient feature seems marginal. This can be found in experimental results in Table, proposed objective (7) gives similar or worse result on L MNIST(similar), CelebA(worse), and Affectnet(worse) dataset. Also, visualization of salient features in Figure 4 show that representation balancing effects of proposed regularization terms are marginal compared to cVAE.

Another cocern is proposed regularization 1,2 might hurt the original ELBO objective. This concern is also related to beta-VAE, the paper showed that large coefficient for prior regularization simply makes the disentanglement effect for VAE. On the other hand, proposed objective (7) requires 'large constant' for approximate quadratic loss. This might induce unintentional negative effects for the objective (7) and objective (8). It would be great if authors can show the proposed regularization does not induce such unintended numerical problem. A possible solution might be showing asymptotic theoretical guarantees as in semi-implicit variational inference.

The objective (8) seems to require additional computational cost to solve linear programming problem. It is not clear if it is effective enough to bear that extra cost; it is hard to conclude that proposed regularization performs better than previous works without confidence interval.

**Experience Assessment:**

I have read many papers in this area.

**Review Assessment: Checking Correctness Of Derivations And Theory:**

I assessed the sensibility of the derivations and theory.

**Review Assessment: Checking Correctness Of Experiments:**

I assessed the sensibility of the experiments.

**Review Assessment: Thoroughness In Paper Reading:**

I read the paper at least twice and used my best judgement in assessing the paper.

---

> ### Author Response · Authors · 2019-11-14
> **Response to Review #3**
>
> Thanks a lot for your helpful feedback. We address your concerns one by one below.
>
>
> “While authors point out that existing formulation in Abid & Zou (2019); Ruiz et al. (2019) ignores about the prior ....... result on L MNIST(similar), CelebA(worse), and Affectnet(worse) dataset.”
>
> We apologize for a typo in Table 1, row `Ours (Eqn. 7)’ for the CelebA dataset, which we corrected. In addition we think the reviewer may have misunderstood this admittedly confusing table: our methods perform best across all datasets. For columns denoted by $z_x$, both lower accuracy and lower SS are better; for columns titled $s_x$, higher accuracy and higher SS are better. This is explained in the caption of Table 1. To clarify this confusion we redesigned this table in the revised version.
>
> Also note: our methods perform better than both baselines despite having fewer neural networks to train and hence fewer parameters. In both [1] and [2], the authors propose a new KL divergence term which is hard to compute. Hence the authors approximate it using the density-ratio trick, which requires a new discriminator neural network that needs to be trained along with the VAE parameters. Thus this adversarial formulation incurs extra computational cost in the form of discriminator training and additional hyper-parameters which may not be easy to specify. In contrast, we propose two principled and well motivated loss terms that are easier to compute, do not require any additional parameters and perform much better than these baselines.
>
>
> “Also, visualization of salient features in Figure 4 show that representation balancing effects of proposed regularization terms are marginal compared to cVAE.”
>
> The following link (https://drive.google.com/file/d/1X5P2Rx7mlF5WwZ5ZXx8bQO74rwD4ZNCT/view?usp=sharing) shows the zoomed in visualization of the salient features from the target dataset. As illustrated in this enlarged figure, target features learned by our approach are visually better separated from each other than the existing works, indicating that the learnt target signal indeed corresponds to the underlying MNIST digits. Moreover, the clustering accuracy quantitatively validates our better performance.
>
>
> “Another concern is proposed regularization 1,2 might hurt the original ELBO objective. This concern is also related to beta-VAE, the paper showed that large coefficient for prior regularization simply makes the disentanglement effect for VAE.”
>
> In the beta-VAE setup [5], more weight is placed on the KL divergence term as compared to the reconstruction loss. However, we do not place any such extra weight on the KL divergence. Instead, we only add positive terms to the RHS of an upper bound on the negative log-likelihood in Equation 6 (e.g., adding the Wasserstein loss term in Equation 8). Thus the upper bound on the negative log-likelihood of the data is still valid with our regularized loss in Equation 8.
>
>
> “On the other hand, proposed objective (7) requires 'large constant' .... does not induce such unintended numerical problem.”
>
> $\lambda_s = 1$ is sufficient in all our experiments. Also, the value of $\lambda_z$ appearing in Equation 8 is set to 1. These values for $\lambda$ don’t induce any negative effects. For reproducibility we submitted code for all experiments.
>
>
> “A possible solution might be showing asymptotic theoretical guarantees as in semi-implicit variational inference.”
>
> Thanks for pointing out this reference, we think [5] is only tangentially related to the contrastive setting. Specifically, we do not assume any distributions on the VAE parameters as done in [5].
>
> “The objective (8) seems to require additional computational cost to solve linear programming problem. It is not clear if it is effective enough to bear that extra cost; it is hard to conclude that proposed regularization performs better than previous works without confidence interval.”
>
> To compute the Wasserstein loss $W_2^2(q_t(z),q_b(z))$, we use mini-batches of samples from both the distributions $q_t(z)$ and  $q_b(z)$. The batch size is 64. Hence we can efficiently compute the optimal matching matrix using standard optimal transport libraries such as [3]. Hence our proposed loss term only incurs a small computational overhead. This is reflected in the running times of our algorithm with and without the loss term $L_{W_2}$:
>
> Time to run 1 epoch:
> Ours (Eqn. 7) [without the term $L_{W_2}$]: 16.9 secs
> Ours (Eqn. 8) [with the term  $L_{W_2}$]: 17.3 secs
>
>
> References:
> [1] Contrastive Variational Autoencoder Enhances Salient Features: https://arxiv.org/abs/1902.04601
> [2] Learning Disentangled Representations with Reference-Based Variational Autoencoders: https://arxiv.org/pdf/1901.08534.pdf
> [3] Python Optimal Transport: https://github.com/rflamary/POT
> [4] Semi-Implicit Variational Inference: https://arxiv.org/pdf/1805.11183.pdf
> [5] beta-VAE: Learning Basic Visual Concepts with a Constrained Variational Framework: https://openreview.net/pdf?id=Sy2fzU9gl

---

### Official Review · AnonReviewer2 · 2019-10-26
**Official Blind Review #2**

**Rating:** 3

**Review:**

This paper is concerned with contrastive disentanglement. The considered problem is interesting and important in the community, and the proposed method seems to be of practical use, according to the empirical results. My concern is that the contribution of the paper on the theoretical or methodological side seems a bit weak.

The proposed method relies on the VAE framework for contrastive disentanglement. The authors proposed two modifications: one is to explicit enforce the posterior of the content representation to be a Delta distribution, and the other is to make sure that the background representation has the same distribution across the background and the target. In the original cVAE framework for contrastive disentanglement, the two constraints were considered in a slightly more implicit way. It is not surprising to see that the proposed method seems to outperform cVAE only slightly. To see how much we can really gain by further explicitly enforcing the two constraints, the authors may do some further studies with varied sample sizes. BTW, in the result for CelebA in Table 1, cVAE seems to outperform the proposed method, denoted by 'Ours', according to Acc of z_x, but it was indicated in the text (as well as in by the bold font) that 'Ours' performed the best--is there a typo?

I acknowledge I read the authors' response and other reviews and would like to keep my original rating.

**Experience Assessment:**

I have read many papers in this area.

**Review Assessment: Checking Correctness Of Derivations And Theory:**

I carefully checked the derivations and theory.

**Review Assessment: Checking Correctness Of Experiments:**

I assessed the sensibility of the experiments.

**Review Assessment: Thoroughness In Paper Reading:**

I read the paper thoroughly.

---

> ### Author Response · Authors · 2019-11-14
> **Response to Review #2**
>
> Thanks a lot for your helpful feedback. We address your concerns one by one below.
>
>
> “The proposed method relies on the VAR framework for contrastive disentanglement.”
>
> We assume this should have read “VAE framework for contrastive disentanglement”?
>
>
> “In the original VAE framework for contrastive disentanglement, the two constraints were considered in a slightly more implicit way.”
>
> If we understand correctly, you mean to refer to cVAE by “original VAE framework for contrastive disentanglement”. If true, we emphasize that the cVAE setup in [1] does not consider the background latent prior to be a Delta distribution at zero in its loss function. The crucial KL divergence term is missing. We highlight this in Section 3.1 of our paper. The assumption about background latent factors being the same is considered implicitly in [1]. However, empirical results for [1] in our Table 1 corresponding to column $L_{W_2}$ suggest that a stronger enforcement is necessary and this improves results significantly from 38.26 to 11.07 in case of CelebA.
>
>
> “It is not surprising to see that the proposed method seems to outperform cVAE only slightly.”
>
> We respectfully disagree with this comment. As highlighted in Table 1 of the paper, our method shows a healthy and consistent 3% improvement on linear MNIST, 5% improvement on non-linear MNIST, 4% improvement on CelebA and 5% improvement on the Affectnet datasets. This clearly demonstrates the superiority of our method over existing state-of-the-art approaches. We obtain this better performance with having fewer neural networks to train and fewer parameters (and hence lower computational cost). We use principled and well motivated loss functions. Hence we believe our qualitative and quantitative results show significant improvements over the state-of-the-art.
>
>
> “To see how much we can really gain by further explicitly enforcing the two constraints, the authors may do some further studies with varied sample sizes.”
>
> Thanks for the suggestion. We have performed additional experiments with varied sample sizes and found that our methods still consistently outperform the existing approach from [1] across a variety of sample sizes. These results are provided in the link (https://drive.google.com/file/d/1vDjfCZ1aulVp1drTMTMik3EZqh77wC1h/view?usp=sharing). The experimental details are as follows: In our experiments, we found that [1] performs the best among all the other baselines as seen from Table 1 of our paper, hence we consider [1] to be our baseline. With regards to varied sample sizes, for the target dataset containing MNIST linearly superimposed on grass (L-MNIST), we chose the number of samples from each digit class to be 1000, 2500 and 4986. Similarly, for the background dataset of just grass images, the total number of samples are 4986, 2500 and 1000 respectively.
>
>
> “BTW, in the result for CelebA in Table 1, cVAE seems to outperform the proposed method, denoted by 'Ours', according to Acc of $z_x$, but it was indicated in the text (as well as in by the bold font) that 'Ours' performed the best--is there a typo?”
>
> We acknowledge that this is indeed a typo. We apologize for the resulting confusion surrounding the accuracy scores in Table 1. We updated our revised table with the correct results. Indeed, as we can now see from Table 1,  “Ours (Eqn. 7)” performs the best in the case of CelebA. Thanks for bringing this up.
>
>
> References:
> [1] Contrastive Variational Autoencoder Enhances Salient Features, https://arxiv.org/abs/1902.04601

---

### Author Response · Authors · 2019-11-14
**General comments**

We would like to thank all the reviewers for their constructive feedback and valuable suggestions. We have now updated our paper taking these comments into account which has helped us make our submission stronger. The following is a summary of updates in our revised version:

1. In our earlier version there was a typo in Table 1 which we have now corrected. Also it seems that there has been some confusion regarding our Table 1 in the paper. To clarify this confusion we redesigned this table. Please note that for columns denoted by $z_x$, both lower clustering accuracy (CA) and lower silhouette score (SS) indicate better performance, whereas for columns titled $s_x$, higher CA and higher SS are better. Thus our method shows a healthy and consistent improvement over the state-of-the-art across all the datasets.

2. We have now added a proof for the fact that our proposed quadratic loss term $E_q[\mu(y)^2+\sigma(y)^2]$ exactly equals the squared second-order Wasserstein distance $W_2^2(q(s|y), \delta\{s=0\} )$ averaged over $y$. We included this as Lemma 2 in the Appendix of our revised paper. We thank Reviewer #1 for this suggestion.

3. We have included new additional experiments with varied sample sizes for background and target datasets (as suggested by Reviewer #2) and found that our methods still consistently outperform the existing approach from [1] across a variety of sample sizes. Appendix A.4 contains these full details.

References:
[1] Contrastive Variational Autoencoder Enhances Salient Features, https://arxiv.org/abs/1902.04601

---

### Decision · Program_Chairs · 2019-12-19

**Decision:**

Reject

**Comment:**

The paper proposes new regularizations on contrastive disentanglement. After reading the author's response,  all the reviewers still think that the contribution is too limited and all agree to reject.